# Analyzing the sensitivity of quantitative 3D MRI of longitudinal relaxation at very low field in Gd-doped phantoms

Danilo de Iure[1☯]*, Allegra Conti[1,2☯], Angelo Galante[3,4,5], Sara Spadone[1], Ingo Hilschenz[1], Massimo Caulo[1,6], Stefano Sensi[1,6], Cosimo Del Gratta[1,6], Stefania Della Penna[1,6]

1 Department of Neuroscience, Imaging and Clinical Sciences, G. D'Annunzio University of Chieti and Pescara, Chieti, CH, Italy, 2 Medical Physics Section, Department of Biomedicine and Prevention, University of Rome Tor Vergata, Rome, Italy, 3 MESVA, Department of Life, Health & Environmental Sciences, L'Aquila University, L'Aquila, AQ, Italy, 4 INFN, National Institute of Nuclear Physics, Gran Sasso National Laboratories, Assergi, L'Aquila, Italy, 5 CNR, SPIN-CNR Institute, Dept. of Physical and Chemical Sciences, L'Aquila, Italy, 6 Institute for Advanced Biomedical Technologies (ITAB), G. D'Annunzio University of Chieti and Pescara, Chieti, CH, Italy

☯ These authors contributed equally to this work.
* danilo.deiure@unich.it

**Data Availability Statement:** Data are available from the Zenodo repository (https://doi.org/10.5281/zenodo.7414467).

## Abstract

### Purpose

Recently, new MRI systems working at magnetic field below 10 mT (Very and Ultra Low Field regime) have been developed, showing improved $T_1$-contrast in projected 2D maps (i.e. images without slice selection). Moving from projected 2D to 3D maps is not trivial due to the low SNR of such devices. This work aimed to demonstrate the ability and the sensitivity of a VLF-MRI scanner operating at 8.9 mT in quantitatively obtaining 3D longitudinal relaxation rate ($R_1$) maps and distinguishing between voxels intensities. We used phantoms consisting of vessels doped with different Gadolinium (Gd)-based Contrast Agent (CA) concentrations, providing a set of various $R_1$ values. As CA, we used a commercial compound (MultiHance®, gadobenate dimeglumine) routinely used in clinical MRI.

### Methods

3D $R_1$ maps and $T_1$-weighted MR images were analysed to identify each vessel. $R_1$ maps were further processed by an automatic clustering analysis to evaluate the sensitivity at the single-voxel level. Results obtained at 8.9 mT were compared with commercial scanners operating at 0.2 T, 1.5 T, and 3 T.

### Results

VLF $R_1$ maps offered a higher sensitivity in distinguishing the different CA concentrations and an improved contrast compared to higher fields. Moreover, the high sensitivity of 3D quantitative VLF-MRI allowed an effective clustering of the 3D map values, assessing their

**Funding:** For S.D.P., S.Sp., I.H. and C.D.G.: This work was carried out under the OXiNEMS project (www.oxinems.eu), funded by the European Union's Horizon 2020 research and innovation program under Grant Agreement No. 828784. For S.D.P., C.D.G., S.Se. This work was also supported by the "Departments of Excellence 2018–2022", MIUR, for the Department of Neuroscience, Imaging and Clinical Sciences, University of Chieti-Pescara. D.d.I. was funded by the project: PON RICERCA e INNOVAZIONE 2014-2020, Dottorati innovativi a caratterizzazione industriale A.A. 2017-2018 - DOT1353282. The funders had no role in study design, data collection and analysis, decision to publish, or preparation of the manuscript.

**Competing interests:** The authors have declared that no competing interests exist.

reliability at the single voxel level. Conversely, in all fields, $T_1$-weighted images were less reliable, even at higher CA concentrations.

## Conclusion

In summary, with few excitations and an isotropic voxel size of 3 mm, VLF-MRI 3D quantitative mapping showed a sensitivity better than 2.7 $s^{-1}$ corresponding to a concentration difference of 0.17 mM of MultiHance in copper sulfate doped water, and improved contrast compared to higher fields. Based on these results, future studies should characterize $R_1$ contrast at VLF, also with other CA, in the living tissues.

## Introduction

In the latest years, Ultra-Low-Field (ULF, 1–100 μT) and Very-Low-Field (VLF, 1–60 mT) MRI raised the interest of the scientific community [1, 2], owing to the intrinsic advantages of the low magnetic field regime. Some of these advantages are: the compatibility with other instruments such as Magnetoencephalography (MEG) systems [3–6]; the possibility of directly imaging neural currents in cerebral regions responding to a stimulus [7, 8]; the possibility of mapping tissues' electric conductivity in brain tissues through current density imaging [9, 10]; artifact-free imaging in presence of metals, thanks to the reduced sensitivity to magnetic susceptibility [11, 12]; easier operation for specific patient populations (children, pregnant women, patients with metallic prostheses/electronic implants) [13, 14]; lower cost compared to High Field (HF, 1–3 T) devices and the possibility to implement bedside/emergency/portable setups with a considerable impact on MRI usage also in rural environments and mobile setups [1, 15]. Notably, ULF and VLF MRI scanners provide improved contrast in $T_1$-weighted images [16] and may represent a promising tool for investigating tumours' heterogeneity even without the use of Contrast Agents (CA), based on the intrinsic differences between the $T_1$ values of healthy and pathologic tissues, and the reduced signal-loss due to magnetic susceptibilities variations [11, 17].

Quantitative MRI (qMRI) provides, from a set of images, numerical information of the physical and chemical properties of tissues, such as the longitudinal and transversal relaxation rates $R_1$ (defined as $1/T_1$) and $R_2$ (defined as $1/T_2$), and the spin density. At high field, this technique was recently boosted thanks to the improvement in acquisition speed (see e.g. methods for MRI fingerprinting [18]). An important aspect of clinical application is the reproducibility of the results, where quantitative MRI has a clear advantage [19]. Interestingly, moving qMRI to lower fields should result in a larger dispersion of the tissues' intrinsic relaxation parameters, thus producing an improved endogenous contrast [17].While fast 3D qMRI at 0.1 T was demonstrated [20], below 10 mT mainly 2D quantitative maps without slice selection (hereafter projected 2D mapping) were reported [2, 4, 16, 17], but the literature on 3D qMRI of longitudinal relaxation at VLF/ ULF is scant, mainly due to the poor SNR at such fields affecting the possibility of applying more time-efficient mapping sequences with a suitable spatial resolution for possible in vivo application.

To evaluate the sensitivity of our VLF MRI scanner operating at 8.9 mT [6] in quantitatively mapping physical and chemical parameters, we imaged phantoms consisting of different concentrations of a commercial Gadolinium-based CA, i.e. MultiHance® (gadobenate dimeglumine, Gd-BOPTA, 0.5 mmol/mL) in water with copper sulfate. The longitudinal relaxation rate $R_1$ linearly depends on the CA concentration and the relaxivity $r_1$. The latter depends on

various parameters [21]. Previous relaxometry studies have demonstrated the $r_1$ dependence of different Gadolinium-chelates on the magnetic field strength from 10 kHz to 300 MHz, including also Gd-BOPTA [22–25]. A general increase of the $r_1$-values in the low frequency region is established, which should reflect in a contrast enhancement of $T_1$-weighted images [26–29]. Thus, using a CA with a known relaxivity allowed us to perform a controlled analysis of the qMRI sensitivity as well as a quantitative comparison of our system and higher field ones.

To have a solid foundation to compare the different systems, we performed quantitative MRI on the same phantoms using CAs for different field strengths. To run this analysis, we compared the contrast in 3D maps of $R_1$ and a set of $T_1$-weighted images, obtained with our VLF MRI (8.9 mT) system and clinical scanners using Low-, Standard- and High- magnetic field strengths (0.2 T, 1.5 T, and 3 T, respectively). Given that, below 20 mT, the $r_1$ of Multi-Hance was shown to be approximately constant [22], at 8.9 mT we should be operating already in the regime of maximum contrast. We thus expect an increased contrast on parametric maps and images. In addition to an analysis distinguishing the single vessels in the phantom, we assessed the sensitivity of VLF qMRI in disentangling different CA concentrations at the single voxel level through an automatic clustering over the 3D phantom maps.

## Methods

### Phantoms

The phantom consisted of 5 plastic vessels (cylindrical tubes with an internal diameter ~9 mm), with identical volumes, filled with about 1.5 ml of different solutions and placed in a piece of foam in a fixed position as depicted in Fig 1. The foam acted as a support, with dimensions fitting into the receiving coils of the MRI systems (minimum diameter ~80 mm). The foam's NMR signal contribution was negligible as far as we were able to tell. Four vessels were filled with different CA dilutions (MultiHance®, Bracco, IT) in doped water (i.e. copper sulphate pentahydrate, $CuSO_4 \cdot 5H_2O$, 1000 ml $H_2O$, 770 mg $CuSO_4$, 1 ml arquad, 0.15 ml $H_2SO_4$, a commercial product by Labochimica srl, 'RAME SOLFATO Sol. Sec. Form. ml 1000'). The fifth tube contained doped water only, without CA, hereafter named the *reference* sample, with relaxation properties closer to brain tissue than pure water.

The four different Gd-dilutions in doped water were 1:3000, 1:2000, 1:1000, 1:500 with respect to the original concentration, corresponding to [0.17, 0.25, 0.5, 1] mM of MultiHance. The four concentrations were chosen to span from the highest one, corresponding to the concentration occurring in plasma right after a bolus injection of CA [30], to the lowest one, similar to the mean Gd- concentration found in muscles [31]. The longitudinal relaxation rate $R_1$ of each sample depends on the Gd- concentration [Gd] according to the formula:

$$R_1 = R_{1,r} + r_{1,Gd} \cdot [\text{Gd}] \tag{1}$$

valid for small concentrations [Gd], where $R_{1,r} = \frac{1}{T_{1,r}}$ is the longitudinal relaxation rate of the reference sample and $r_{1,Gd}$ is the CA relaxivity.

### Recordings

MR images were recorded using four devices operating at different magnetic field strengths: a system implemented in the EU project MEGMRI, operating at 8.9 mT [6], and three commercial systems used for clinical applications operating at 0.2 T (Esaote Artoscan C-Scan), 1.5 T and 3T (both Philips Achieva). The VLF system operated at a magnetic field of 8.9 mT generated by a compensated solenoid coil. A Maxwell coil generated a gradient field of 0.22 mT/

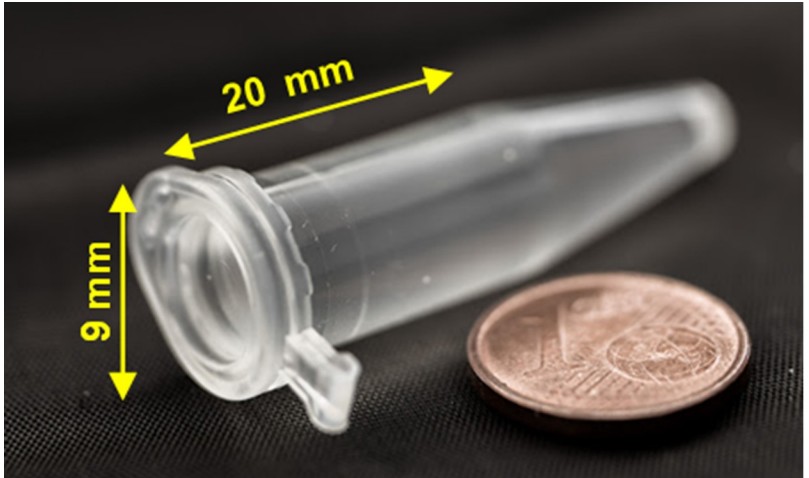
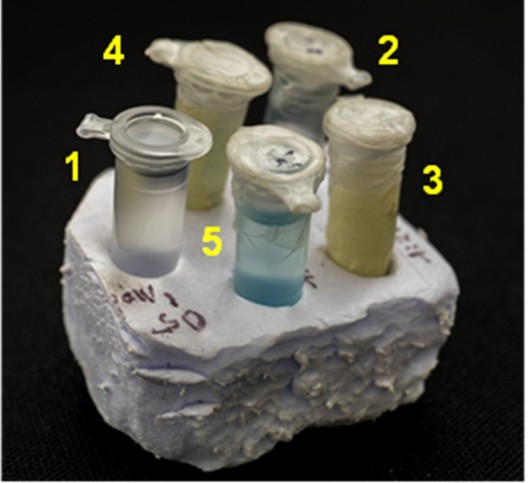
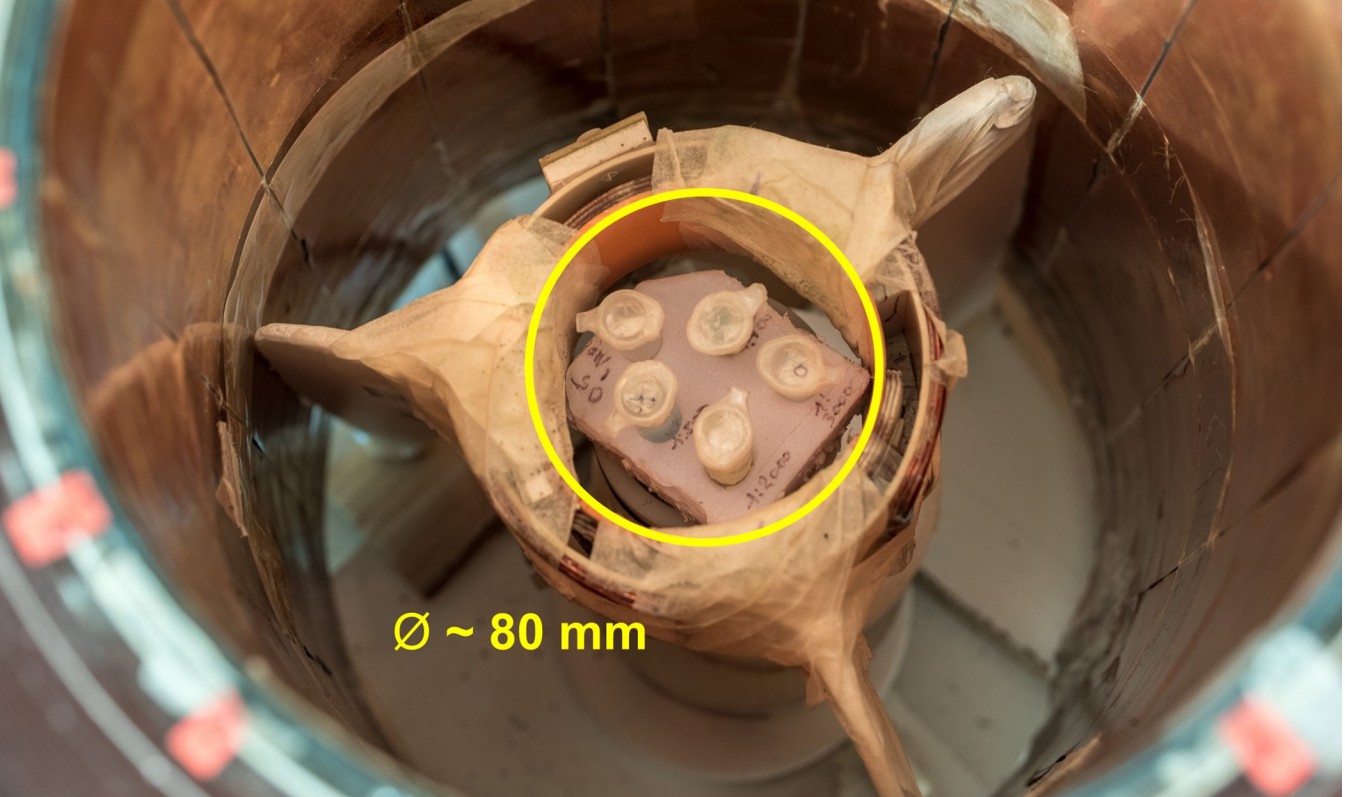

**Fig 1. *Upper left*: An individual phantom vessel.** *Upper right*: The 5 samples in the foam support, same positions as for the measurements. The samples are positioned as follows: 1- ref. sample, 2–0.17 mM, 3–0.25 mM, 4–0.5 mM, 5–1 mM. Each vessel is filled with about 1.5 ml of solution. *Lower*: The phantom is shown inside the VLF system.

(m·A) in the Z-direction (along the solenoid axis) and the X-Y gradient coils were designed using a Finite Element method and were located on the inner surface of the solenoid generating a gradient field of 0.38 mT/(m·A). The slew-rate was approximately 5 T/(m·s). The receiver coil $R_x$ was a saddle style coil with a diameter of 8 cm and height of 6 cm. The transmission coil $T_x$ was placed outside the $R_x$ coil, rotated by 90° to achieve a decoupling of around 60 dB. The Artoscan was equipped with a permanent magnet to generate the 0.2 T field and field

**Table 1. MRI acquisition parameters.**

|  | 8.9 mT | 0.2 T | 1.5 T | 3 T |
|---|---|---|---|---|
| Sequence | Spin-Echo 3D | Multislice Spin-Echo | Multislice Spin-Echo | Multislice Spin-Echo |
| $T_E$ (ms) | 19 | 18 | 8 | 12 |
| $T_R$ (ms) | [85, 100, 110, 120, 170, 250, 340, 510] | [80, 160, 240, 340, 440, 500, 740, 1000] | [160, 240, 340, 440, 500, 740, 1000] | [100, 160, 240, 340, 440, 500, 740, 1000] |
| **FA (°)** | 90 | 90 | 70 | 70 |
| **Phase encoding steps** | 32x32 | 136 | 148 | 152 |
| **Frequency encoding steps** | 32 | 192 | 152 | 152 |
| Voxel size (mm³) | 3.0x3.0x3.0 | 0.5x0.5x3.0 | 0.9x0.9x3.0 | 1.0x1.0x3.0 |
| Min/Max Acquisition times (s) | [348, . . ., 2089]* | [220, . . ., 540] | [68, . . ., 408] | [96, . . ., 612] |
| **Normalization** | - | Mean noise | - | STD of noise |

Echo Time ($T_E$), Recovery Time ($T_R$), Flip Angle (FA), spatial resolution and acquisition times at different magnetic fields. The slices in the multislice case are adjacent to each other. Noise mean and standard deviation (std) for the normalization are derived from all the voxels external to the borders identified with the segmentation algorithm.

* Please note that the 8.9 mT 3D cartesian spin-echo is not optimized, since 32 slices have to be acquired, regardless of the phantom dimensions. Conversely, at higher fields 2D multislice sequences are used.

gradients of 10 mT/m with a slew rate of about 40 T/(m·s). This system was designed to image upper and lower limbs. The Philips Achieva systems were equipped with superconducting magnets and gradients of 33 mT/m and 40mT/m and slew rates 180T/(m·s) and 200 T/(m·s), respectively.

At VLF, $T_1$-weighted images were acquired using an isotropic 3D cartesian Spin-Echo sequence, with two phase encoding gradients and no k-space subsampling. At higher fields a cartesian multislice Spin-Echo was used with standard clinical settings, ensuring that the slice order acquisition was alternated to minimize magnetic transfer effects between adjacent slices. The k-space was under-sampled to optimize acquisition performances (according to the default suggested value from the commercial software, see Table 1) and spoiling gradients were applied at the end of each repetition to avoid spurious signal in the next one.

For each scanner, the image volume was centred, along the longitudinal direction, on the common vessels centre, and only the central slices were considered for the subsequent analysis, to avoid any edge effect contribution. At each field, images were collected for different ranges of Repetition Times ($T_R$), selected to span the tube relaxation functions up to at least twice the $T_1$ value of the reference sample. The Echo Time ($T_E$) was fixed at the smallest value permitted by each scanner, to minimize effects of the transversal relaxation on the signal magnitude. Finally, in each scanner, the same number of Excitations (NEX) was used, namely 4. All the acquisition parameters are summarized in Table 1.

## Contrast analysis on $R_1$ maps

The data were preprocessed as reported in the S1 File (see *Data Preprocessing* subsection). Notably, the voxel size in all the images recorded with the commercial scanners was rescaled to the VLF system. Then $R_1$ was estimated at each voxel by modelling and fitting the voxel signals in the masked images with the following $T_R$ function [32]

$$S(T_R) = A \cdot \left(1 - e^{-T_R/T_1}\right) \tag{2}$$

where $A = S_0 \cdot e^{-T_E/T_2}$ is a constant depending on the transversal relaxation process, $T_E$ is

specific for each scanner, the signal amplification was kept fixed for each set of acquisitions, and the spin density is $S_0$. In this study, $T_E$ was the lower limit allowed by each system and thus varies across systems. Eq 2 is a simplified version of a more complex formula found for repeated SE sequences [33], valid for $T_E/2 \ll T_R$ (which was always true, see Table 1) and $T_2^* \ll T_R$ [34].

The latter condition is met in our VLF setup where, during the acquisition window, the $T_2^*$ is mainly driven by the gradient field. In fact, according to [35], field inhomogeneities affect the $T_2^*$ as in the following:

$$\frac{1}{T_2^*} = \frac{1}{T_2} + \gamma \Delta B_{inhom} \tag{3}$$

where $T_2$ is the intrinsic transverse relaxation time, and $\Delta B_{inhom}$ is the magnetic field inhomogeneity across a voxel. To obtain an upper estimate of $T_2^*$ we consider only the applied readout gradient discarding contributions from the intrinsic relaxation and the measurement field inhomogeneity. The applied sequence bandwidth is about 2.5 kHz, corresponding to 78 Hz for each of the 32 acquired voxels in the frequency direction. This results in a $T_2^* \leq 1/\gamma \Delta B_{inhom} = 13$ ms constraint, confirming the $T_2^* \ll T_R$ assumption (as the smallest $T_R$ used in our procedure was 85 ms). The validity of this assumption was also confirmed by inspecting the signal amplitudes at the end of the acquisition window too, as reported in S1 File (see the sub-section *Control measurement on the assumption $T_2^* \ll T_R$*), further solidifying the approximation.

The same held true at higher fields as well, where nulling of transverse magnetization before the excitation pulses was achieved by the inclusion of spoiling gradients. Then, for each field strength, the following steps were applied to evaluate the sensitivity to different $R_1$ values at the vessel level equally across different systems:

i.  $R_1$ values of all voxels in each vessel were analysed by 1-way ANOVA with the measurement field as a categorical factor, to evaluate possible effects of both CA concentration and field. Bonferroni's post-hoc analysis was applied to significant ($P<0.05$) effects.

ii.  The differences between all the possible $R_1$ map voxel pairs of two samples with contiguous dilutions (absolute contrast) were computed for every measurement field.

iii.  The $R_1$ contrast values across vessel pairs were compared through 1-way ANOVA analysis over all possible voxel differences with the measurement field as a categorical factor, to evaluate possible effects of the vessel pair and the measurement field. Bonferroni's post-hoc analysis was applied to significant effects.

## Contrast analysis on $T_1$-weighted images

We also compared the sensitivity of the systems with spin-echo $T_1$-weighted images. To account for the different $T_1$ values of the reference sample for each field, we selected images obtained at specific repetition times $T_R^*$. We first estimated the $T_1$ values at each field, by fitting each voxel, within the sample mask, with the function in Eq 2, and then averaged over the sample. For each measurement field, $T_R^*$ was selected as the closest among the available $T_R$ values, to the estimated $T_1$ of the reference sample. We then quantified the contrast in the $T_1$-weighted images between vessel pairs using the following procedure:

i.  For each field strength, the images acquired at $T_R^*$ were rescaled in a 0–255 scale dividing all the voxel values by the factor *max(voxels)*/255;

ii. points (ii) to (iii) from the previous subsection *Contrast Analysis on $R_1$ maps* were applied to the rescaled intensities of $T_1$-weighted images, to evaluate pairwise contrast over the vessels in the phantom.

## Clustering analysis

An automatic clustering procedure was used to determine whether, despite the low available SNR, the sensitivity of the VLF system was suitable to automatically distinguish individual voxels in the $R_1$ maps for different CA concentrations. The clustering procedure assigns homologous voxels to the same group, and we applied it to assess whether all the voxel in a vessel were grouped together. Voxels of the $R_1$ map and within the mask were clustered through a K-means procedure using a variable number of clusters (from two to nine). The optimal number of clusters (which is expected to be five) was automatically estimated through the Krzanowski-Lai criterion and maximization of the product of different clustering performance criteria [36–39]. Clustering results were evaluated in terms of accuracy in grouping voxels with the same CA concentration and in separating voxels corresponding to different CA concentrations. Specifically, for each vessel we computed the within-sample homogeneity, defined as the maximum over clusters, of the percentage of voxels in each vessel belonging to the same cluster. The percentage was referred to the total voxel number of the related vessel. Moreover, we estimated the across-samples homogeneity, defined as the percentage size of the most representative cluster of each vessel, normalized with respect to the total voxel number of the vessel, minus the related within-sample homogeneity. The last measure quantified the assignments outside the sample and in an optimal situation should be 0. Both measures were averaged across the five vessels.

## Results

### 3D imaging of phantom at different field strengths

An example of $T_1$ weighted images recorded at 8.9 mT and 1.5 T is shown in Fig 2 for different $T_R$ values. The images show the phantom comprising the reference vessel and the 4 vessels with different dilutions of MultiHance. With the same number of averages, the SNR and spatial resolution of the images recorded at VLF are lower than at higher fields. Nevertheless, it is

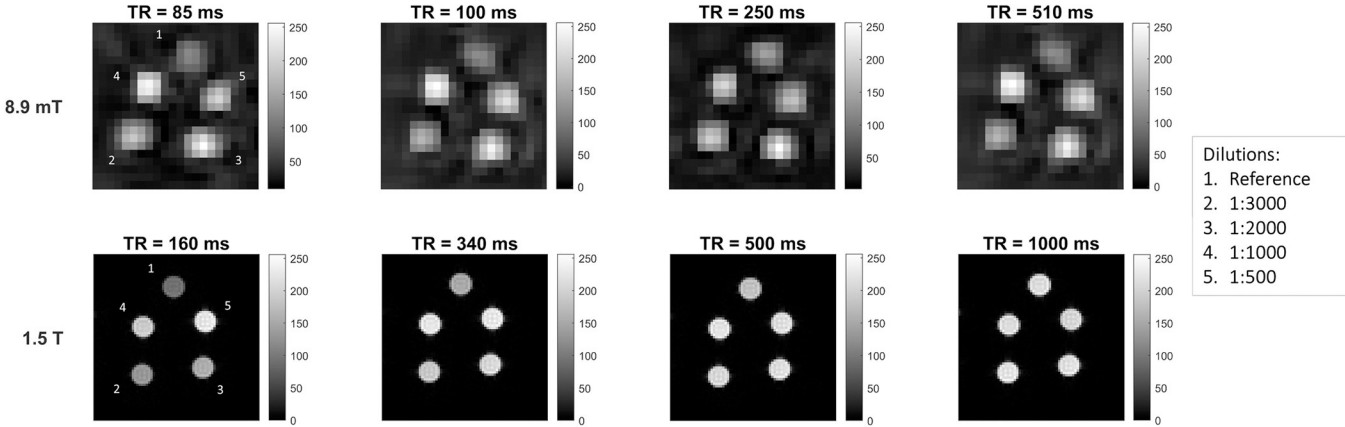

**Fig 2. $T_1$-weighted images of the phantom.** Images, at different $T_R$, of a central slice of a phantom composed by vessels with different MultiHance dilutions. *Upper panel*: MR images obtained at 8.9 mT. *Lower panel*: as a comparison, MR images obtained at 1.5 T.

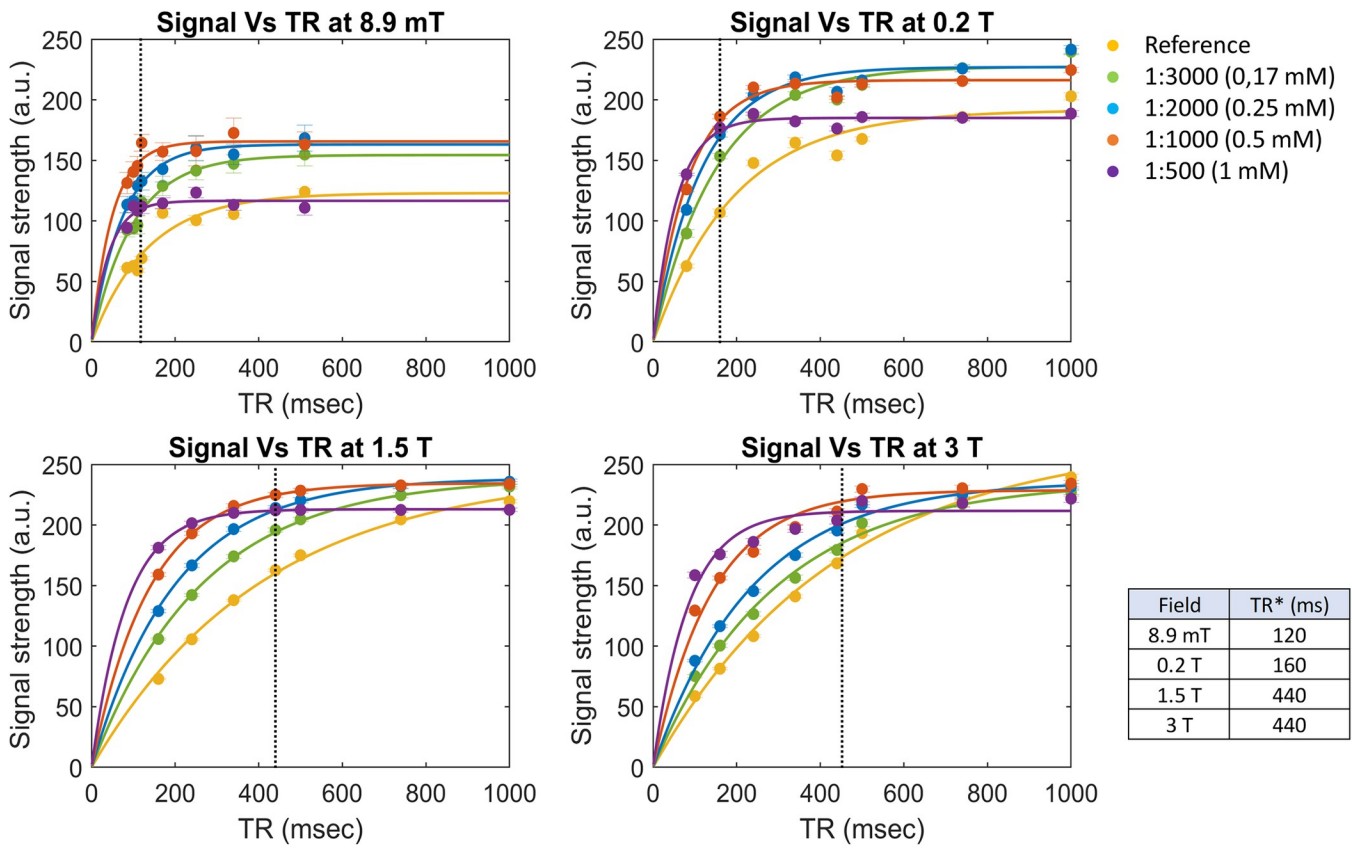

**Fig 3. Mean signal over the samples.** Mean signal from all the voxels of the samples at different $T_R$ and different fields, together with the exponential fitting curve. Signals at 0.2 T and 3 T were normalized to the background noise as reported in Table *1*. Error bars represent the standard error on the mean estimation. The $T_R^*$ chosen for each field is indicated with the dotted line in each plot and reported in Table *1* for each field.

still possible to identify the tubes and the differences in voxel intensities among the different CA concentrations, at least for low $T_R$ values.

In Fig 3 we show the mean signals obtained from the central slices together with the related exponential fits according to Eq 2, we further note that in addition to the $T_1$ contribution, the $T_2$ dependence of the constant A (see Eq 2) influences the asymptotic signal values for different CA concentrations. The experimental $T_{1,r}$ values for the reference sample were about (133±4) ms, (187±2) ms, (413±2) ms, and (445±10) ms at 8.9 mT, 0.2 T, 1.5 T, and 3 T respectively (average over voxels ± standard error). To run a quantitative comparison on image contrast, we selected the optimal $T_R^*$ for each magnetic field among the available $T_R$ values. The closest ones to the experimental $T_1$ of the reference sample were [120, 160, 440, 440] ms.

### Contrast at different field strengths

$T_1$-weighted images of the phantom central slice obtained at the appropriate $T_R^*$ value for each field strength are shown in Fig 4A. Qualitatively, the difference between the reference sample and the vessels containing the Gd dilutions is clearly visible in all fields, whereas differences among the CA concentrations become more evident towards lower fields.

Fig 4B shows the corresponding $R_1$-maps obtained from the masked images, including the edge voxels. In Fig 4C the same slices as in Fig 4B are shown, but after voxel resizing and edge removal for the quantitative analyses. Vessel edges were removed to avoid partial volume

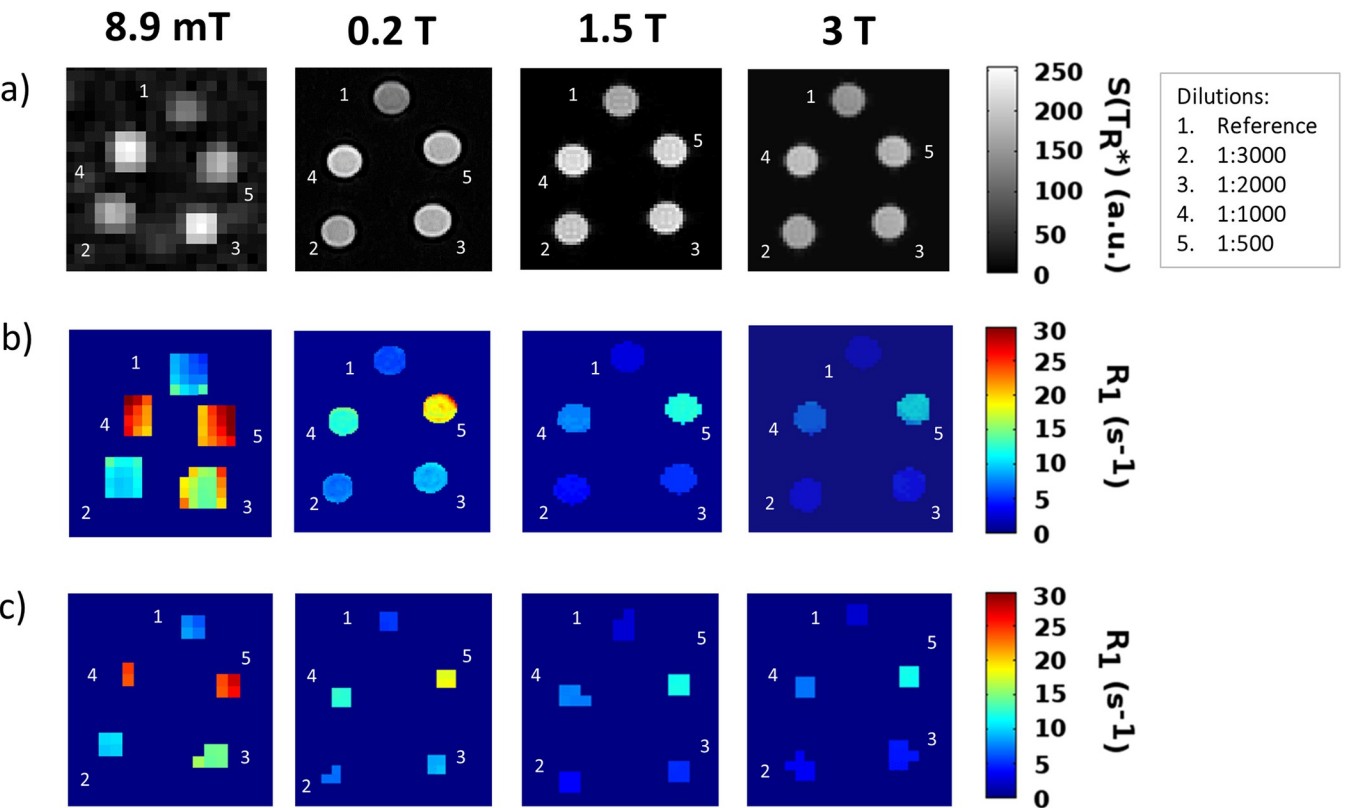

**Fig 4. $R_1$ maps of the phantom.** (a) $T_1$-weighted images (central slice, images at 0.2 T, 1.5 T, 3 T not resized) of the MultiHance phantom, at 8.9 mT, 0.2 T, 1.5 T and 3 T at $T_R = T_R^* = [120, 160, 440,440]$ ms respectively. (b) $R_1$-maps for the same slice obtained at the same fields (no resize) by fitting the voxel signals in a mask containing the edges, as a function of $T_R$ with the Eq 2. (c) $R_1$-maps from the resized images after application of the mask excluding the border voxels. These are sample slices of the actual 3D maps where the contrast and clustering analyses were performed.

effects, which would include the contribution of the background noise. In the VLF regime and to a lesser extent in the other applied fields, the contrast in $R_1$ maps appears larger than in $T_1$-weighted images.

The mean $R_1$ values extracted from these maps for each vessel are summarized in Fig 5, where we can see the increase of the $R_1$ values for all the samples with decreasing field, peaking at VLF. This observation is supported by the literature reporting a longitudinal relaxivity increase at lower magnetic fields in different media [22, 24]. To quantitatively compare the $R_1$ values, we ran ANOVA (see S1 File for details on ANOVA results) which suggested that the vessels could be distinguished, $R_1$ being modulated with the measurement field as well as the CA concentration. Specifically, $R_1$ values at 8.9 mT were significantly larger than for the other fields and the same applies to 0.2 T versus the higher fields, while no significant differences were found between 1.5 T and 3 T. Moreover, for the VLF and the 1.5 T measurements the different vessels showed significantly different $R_1$ values, while it was possible to distinguish only the 3 higher CA concentrations at 0.2 T and 3 T. Finally, we fitted the estimated $R_1$ values reported in Fig 5, using Eq 1, obtaining the estimates of the MultiHance $r_1$ in doped water reported in Table 2.

## Comparison between $R_1$ and $T_1$-w sensitivity at the vessel level

To highlight the improved sensitivity obtained with VLF-quantitative MRI of CA-doped vessels, we analysed through separate ANOVAs (see S1 File for details on ANOVA results) both

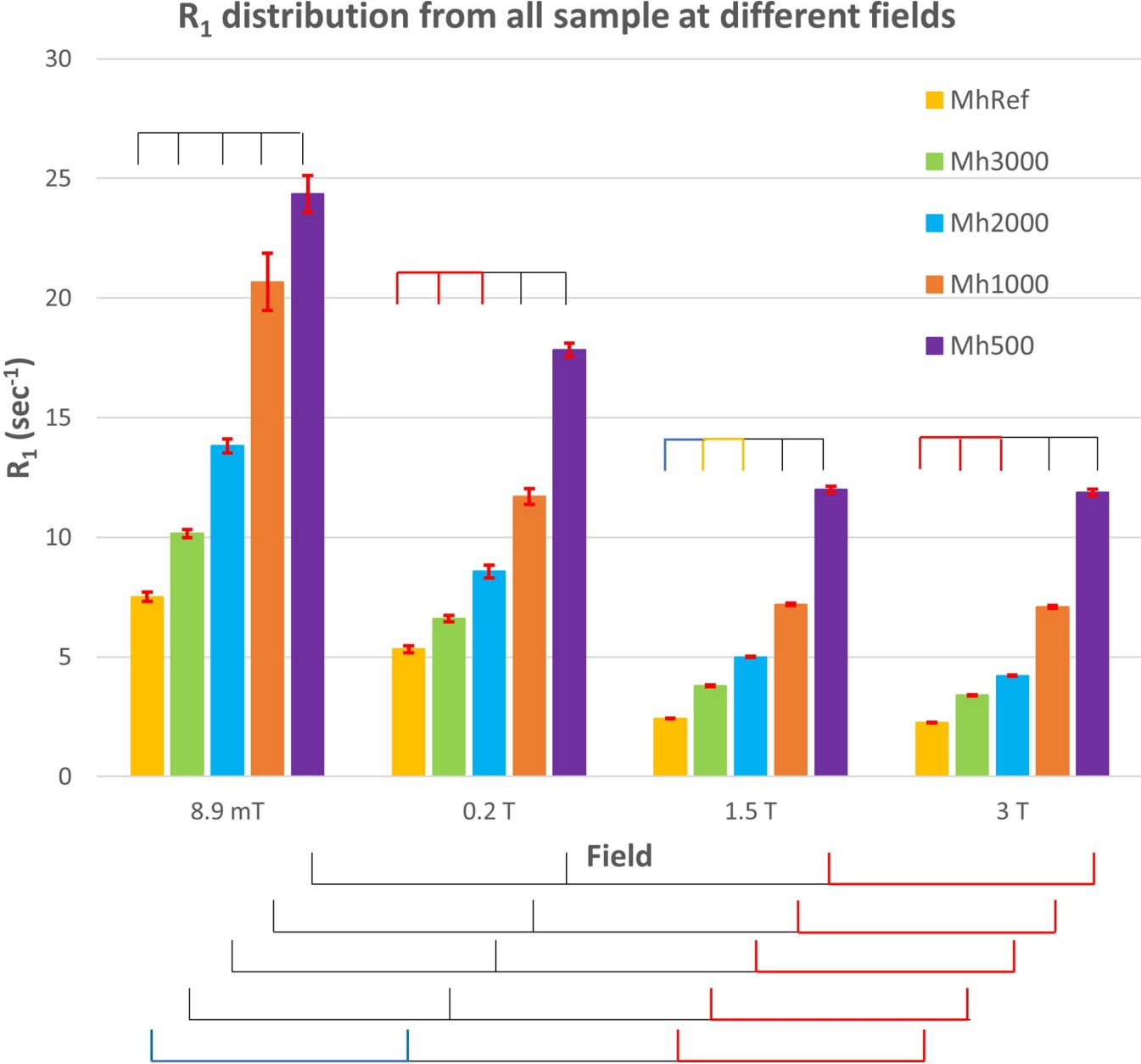

**Fig 5. Histogram of the $R_1$ values.** Mean $R_1$ values for each concentration and main field value with relative standard error as error bars. The mean $R_1$ is the average of the $R_1$ fitted on the single voxel of the images. Results from a 1-way ANOVA with field as categorical factor and Bonferroni correction comparison is also shown: i) spiked segments on top, over the samples at the same field indicate significant (yellow–$P < 0.03$, cyan–$P < 0.003$, black–$P < 3 \cdot 10^{-7}$) and not significant (red) comparisons over subsequent CA concentrations. The reference sample is compared to the lower CA concentration. ii) spiked segments in the bottom indicate significant and not significant comparisons (same notation as above) of the same sample tube across the scanners.

the differences between $R_1$ values ($R_1$ contrast) and then, as a comparison, the ones between the normalized signals (signal contrast) obtained from the phantom vessels across the measurement fields. The normalization of the $T_1$-weighted images on the 0–255 scale allowed a direct comparison of performances at different field strengths.

The bar plot in Fig 6 shows the inter-vessels spreads of the $R_1$ (Fig 6 *left*) and the normalized $T_1$-weighted variations (Fig 6 *right*) between the higher Gd concentration (Mh500) and the reference sample for all magnetic field strengths.

**Table 2. r$_1$ relaxivities.**

|  | 8.9 mT | 0.2 T | 1.5 T | 3T |
|---|---|---|---|---|
| MultiHance r$_1$ (mM· s)$^{-1}$ | 17.3 | 12.7 | 9.6 | 9.8 |
| Fit error (mM· s)$^{-1}$ | 3 | 0.4 | 0.1 | 0.3 |

Relaxivity of MultiHance in water doped with copper sulfate and the related estimation error.

ANOVA confirmed that R$_1$ contrast is significantly larger as the field decreases. Notably, such an increase is not that clear in the T$_1$-weighted image differences, where it seems that the maximum contrast is obtained at 0.2 T, and then decreases with the increasing field. These results suggest that R$_1$ maps at VLF are more reliable than T$_1$-weighted images, due to their higher SNR and the lack of T$_2$ effects which instead affect the T$_1$-weighted images.

In Fig 7 *upper* we show the R$_1$ contrast between pairs of samples with contiguous Gd dilutions, together with the normalized T$_1$-weighted voxel value contrast (Fig 7 *lower*). Qualitatively, the R$_1$ contrast is always positive, in all fields. However, from the ANOVA interaction reported in Fig 5, only at 8.9 mT and 1.5 T all the R$_1$ contrasts were significantly different from zero, while only contrasts between higher concentrations were different from zero at 0.2 T and 3 T. In general, the R$_1$ contrast seems to increase as the measurement field decreases, as confirmed by 1-way ANOVA with the field as the categorical factor, and no significant differences were found between 1.5 T and 3 T. R$_1$ contrast was also significantly larger as the CA concentration increased, and this increase was enhanced by the decreasing field, except for the highest CA concentration at VLF. This effect was possibly induced by inadequate sampling in the low T$_R$ range (see Discussion). Notably, in the other fields, the contrast of Mh3000 vs the reference sample and Mh2000 vs Mh3000 did not differ as at VLF (we even note a significant decrease at 3 T), suggesting an increased sensitivity in the VLF setup to differences in low CA concentrations, consistently with results shown in Fig 5. The contrast at higher CA concentration is instead clearly detectable. When comparing the contrast across the fields we obtained that, except for the contrast between the higher CA concentrations (Mh500-Mh1000), all the other

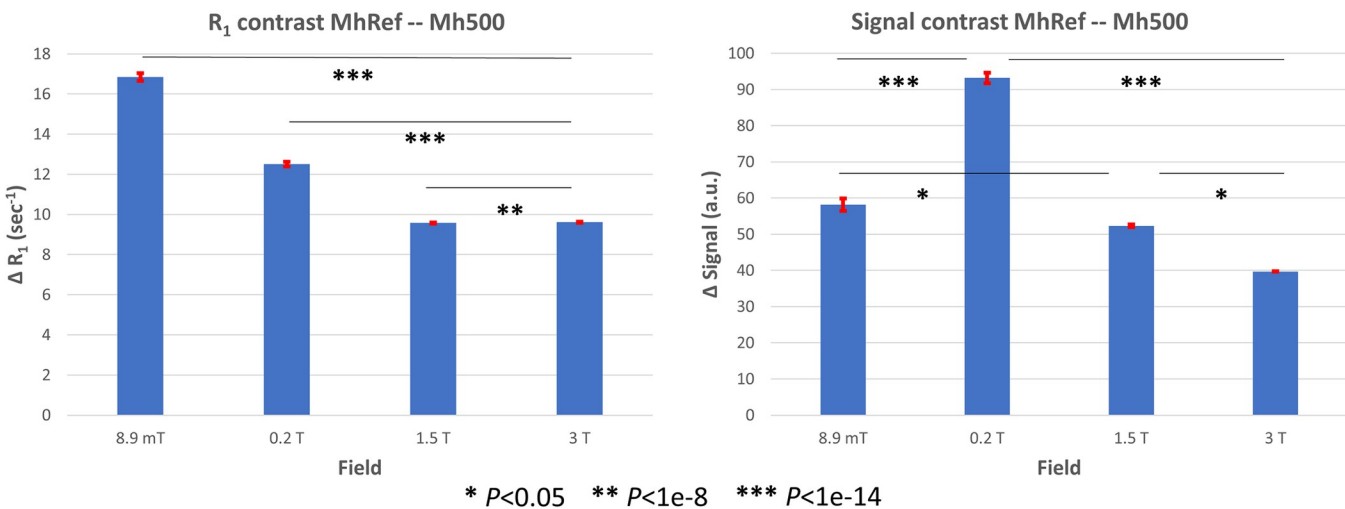

**Fig 6. Contrasts between the extremal dilutions.** *Left*: Relaxation rate contrast between the reference sample and the sample with the lowest dilution (1:500); *Right*: Analogous analysis to the previous one but on the distributions of the differences between the normalized voxel intensities over the T$_1$-weighted images. Significant comparisons are reported on the bar plot. For R$_1$ values, contrast is strongly enhanced at VLF. For images, the contrast enhancement is not clear, probably due to the poor SNR in the VLF images and to T$_2$ effects.

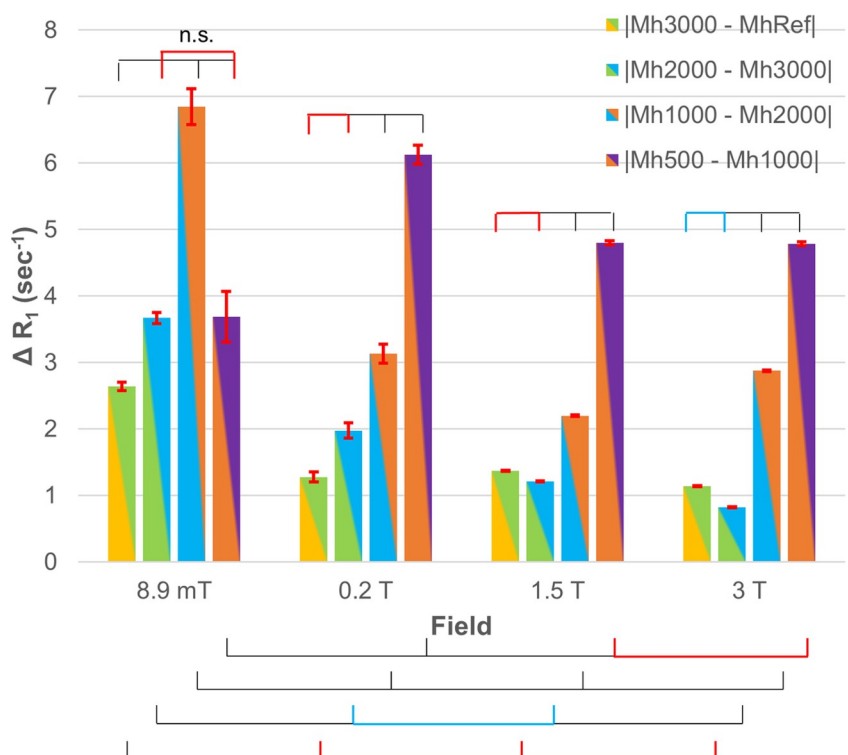

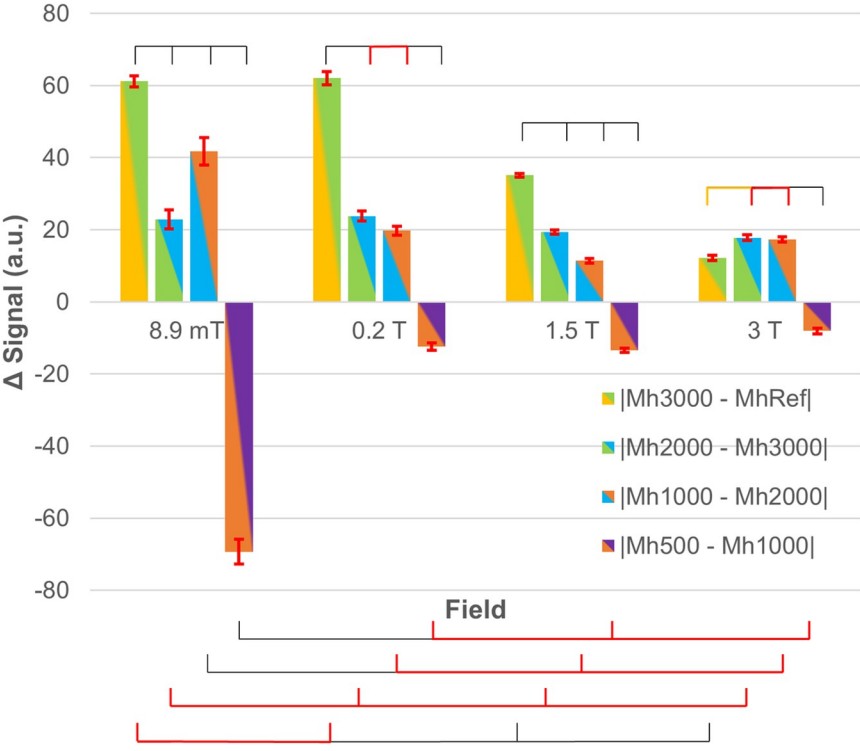

**Fig 7. Contrasts between close dilutions.** *Upper*: $R_1$ contrast between samples with contiguous CA dilutions at different fields. The lower dilution was compared to the reference sample. *Lower*: $T_1$-weighted normalized signal contrast, for $T_R = T_R^*$ at each field, between the same sample pairs. The contrast between the higher CA concentrations is negative at all the measurement fields and is larger at 8.9 mT possibly due to $T_2$ effects enhanced at VLF.

contrasts were always larger at VLF than at the other fields (see Discussion). In summary, with NEX = 4 at VLF we could clearly detect the smaller contrast in our phantom, which was 2.7 s$^{-1}$, corresponding to a concentration of 0.17 mM of gadobenate dimeglumine in copper sulfate doped water.

As a comparison, we show in Fig 7 *lower* the normalized $T_1$-weighted signal contrast estimated at the $T_R^*$ for each field. First, we note that the contrast between the 2 higher CA concentrations is negative (it is significantly different from zero only at VLF), but this, as further discussed later, is due to $T_2$ effects. At VLF, all the other contrasts are significantly different from zero, whereas only the first contrast is significantly different from zero at 0.2 T and 1.5 T and none at 3 T.

The mean $R_1$ (signal) contrast is the average of all the possible $R_1$ (signal) differences over single voxels pairs comprised in the mask and included in two samples with contiguous CA dilutions. Results from a 1-way ANOVA with field as categorical factor and Bonferroni correction comparison is also shown: i) spiked segments on top, over the samples at the same field indicate significant (orange–$P < 0.03$, cyan–$P < 0.003$, black–$P < 3 \cdot 10^{-7}$) and not significant (red) comparisons over subsequent contrasts; ii) spiked segments in the bottom indicate significant and not significant comparisons (same notation as above) of the same sample tube across the scanners.

## Clustering analysis

Finally, we inspected the sensitivity of VLF $R_1$ 3D mapping for single voxels, using a clustering algorithm to automatically assign each voxel to a vessel. The results of the clustering procedure are shown in Fig 8. Notably, the algorithm automatically identified five clusters. Four clusters identified four different vessels (the reference sample–yellow–, the two lower CA concentrations–green and cyan–and the higher CA concentration–violet) except for a few edge voxels. The identification of the Mh1000 sample was not as clear as for the other samples, instead this sample appeared split into two clusters in the upper and lower part along the axial axis. The voxels in the upper part were assigned to the violet cluster, whereas the ones in the lower part were assigned to the fifth cluster, the orange one. The average within-samples homogeneity was 87.5%. Moreover, the grouping error, i.e. the assignments outside the sample quantified through the corresponding average across-sample homogeneity, was 11.7%. These values of the within-samples and the across-samples homogeneity estimated over all the clusters confirmed the capability of VLF $R_1$ maps to distinguish voxels corresponding to different CA concentrations. For the Mh1000 sample, the accuracy decreased to 57%, while the across-sample homogeneity for the violet and orange cluster was 23% and 28% respectively. For this sample, the grouping error is induced by the higher dispersion of the $R_1$ values. The average across-sample homogeneity was also affected by some edge voxels in other clusters, due to the partial volume effect (see Discussion), induced by the large voxel size. Overall, these results confirm an $R_1$ sensitivity value better than 2.7 s$^{-1}$, even at the single voxel level with 3 mm side.

## Discussion

In this work, we evaluated the sensitivity of $R_1$ maps and 3D $T_1$-weighted images obtained with an in-house MRI setup operating at 8.9 mT using a phantom consisting of different CA

# Clustering on the 3D volume

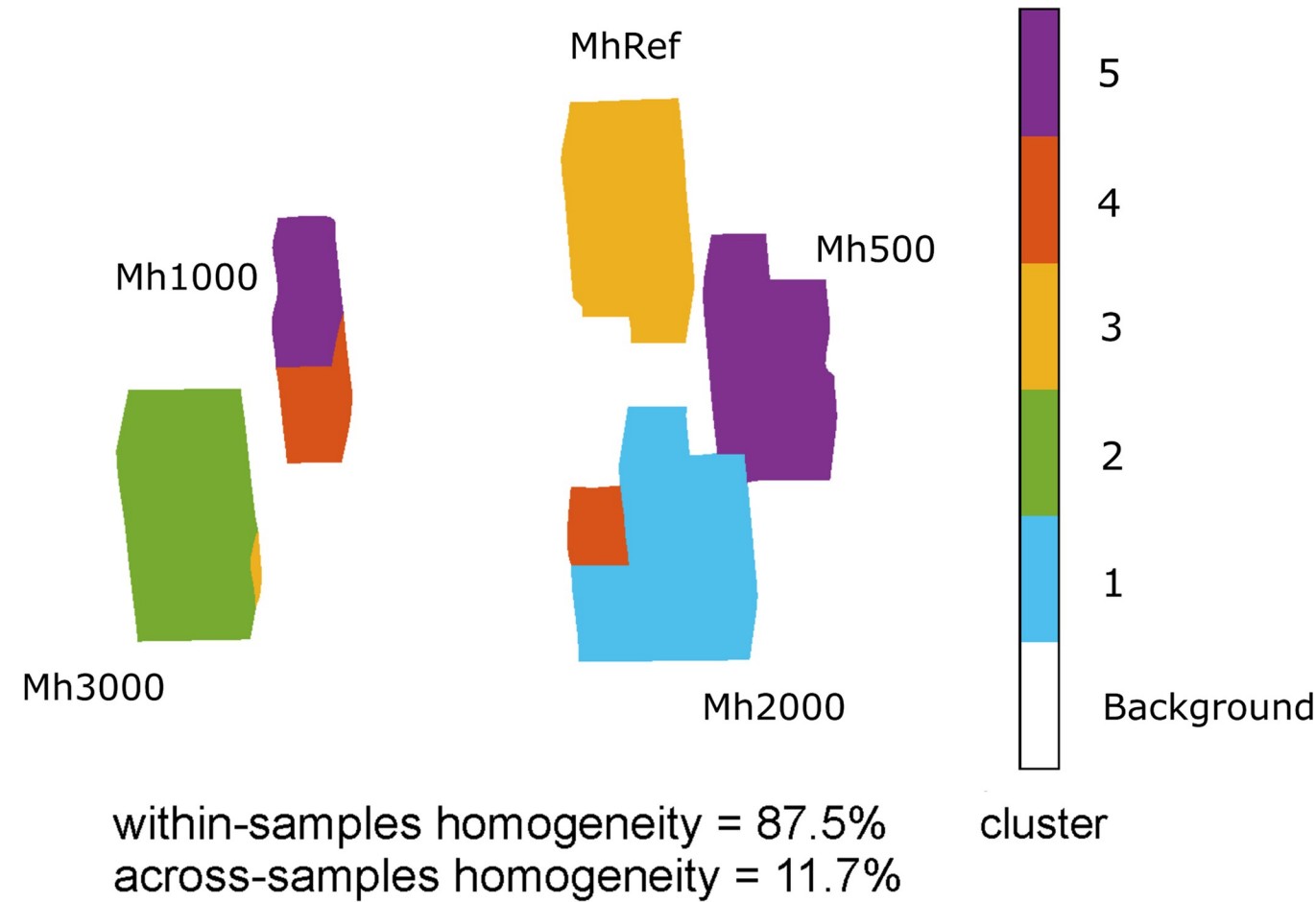

within-samples homogeneity = 87.5%        cluster
across-samples homogeneity = 11.7%

**Fig 8. Clustering results on the 8.9 mT $R_1$ maps.** Optimal clusters number are obtained through a mixed performance criterium (see Methods). Cluster colour and numbering are consistent across clustering types. Homogeneity values reported below each cluster show that voxels are in general grouped according to the CA dilution. Grouping over the two samples at higher CA concentration is less homogenous, due to effects of inadequate sampling at low $T_R$.

concentrations in doped water. For comparison, similar results were also obtained by three other commercial scanners operating at higher fields and used for routine clinical investigation, a system at 0.2 T, 1.5 T, and 3 T. We quantitatively analysed the $R_1$ maps, demonstrating that at VLF we could distinguish the samples and even the voxels at different CA concentrations (as assessed through the automatic clustering). Conversely, the lower CA dilutions could not be separated at higher fields. Moreover, the contrast in $R_1$ maps was, in general, higher at VLF than at the other fields, and increased with the CA concentration, while unclear contrast patterns were obtained when inspecting contrast on normalized $T_1$-weighted images. Overall, we assessed the possibility of quantitative 3D mapping at VLF, able to distinguish different CA concentrations in a clinical-relevant interval, even with the low SNR and spatial resolution at this field, and to provide more sensitive and clear results than $T_1$-weighted image contrast, at least at the sample or voxel group level.

## Methodological considerations on $R_1$ mapping of CA concentration at VLF

A clear result obtained with our VLF MRI system is that the sensitivity of the instrument is suitable to distinguish different CA concentrations in doped water (see Fig 5), even for single voxels which were grouped consistently within the sample they belonged to (see Fig 8). On one side we expect that $R_1$ increases for all the concentrations when lowering the measurement field, thus making it easier to distinguish the different concentrations. Relaxometry measurements on commonly used Gd-based contrast agents showed $r_1$ values higher at ULF/VLF than at higher field strengths [22, 40]. For MultiHance in water, the transition from low to high values occurs in the frequency interval from 5 to 20 MHz, which is above the resonance frequency of our VLF system (375 kHz), includes the 0.2 T scanner and is below the frequencies of the other two scanners. Although this explains why we expect $R_1$ maps at VLF to be more sensitive than in larger fields, ULF/VLF still suffers from poor SNR and limited voxel resolution, and its ability to distinguish different CA concentrations was a challenge. However, $R_1$ maps were more robust than if utilising the signal intensity images. This increases the reliability of $R_1$ maps, as assessed by the clustering results, where even single voxels were correctly assigned to the related sample. Given that $T_1$ of biological tissues should scale with the measurement field (see [17]), increasing contrast, these results are promising in the perspective of in vivo $R_1$ mapping at VLF, also supported by CAs administration.

Instead, interpreting 3D $T_1$-weighted images is more difficult. First, these images suffer from the background noise, which is higher at lower fields and when a small number of averages is used to speed up the recordings. Second, the reduction of $T_2$ at the higher CA concentrations affects the signal intensity and in particular its asymptotic values (see Fig 3 and the coefficient A in Eq 2), making it difficult to interpret the $T_1$-weighted images as in the case of Mh500-Mh1000 in Fig 7 *lower*, where the contrast between these two samples changes polarity compared to the other contrasts. Similar effects are also shown in Fig 6 *right*, where a clear trend of image contrast as a function of the measurement field is not found, even when the highest concentration and the reference sample are compared. Notably, $R_1$ maps do not suffer from these confounds and thus the contrast trend is in general clearer, with an overall increase at VLF compared to the other fields together with an increase with the CA concentration.

## Limitations of $R_1$ mapping at VLF

We have to acknowledge a few limitations for $R_1$ mapping at VLF, which however could be solved in future implementations. The first limitation deals with effects of large $R_1$ and explains the results obtained at VLF on $R_1$ contrast for the higher CA concentrations, and consequently on the clustering at the single voxel level. Specifically, despite the Mh500-Mh1000 $R_1$ difference being significantly larger than zero, it was not as large as expected from the other contrasts trend, while the Mh1000-Mh2000 was larger than expected. This is due to an increase of the error in the estimation of $R_1$ possibly affecting the higher CA concentrations, which was unavoidable with our VLF MRI spectrometer, since the current console did not allow to use $T_R$ values below 85 ms. This led to a sub-optimal sampling for the higher CA concentrations, where $T_1$ was considerably shortened, thus increasing the fit error. This effect also explains the suboptimal grouping for the sample Mh1000, in the 3D clustering (the orange/violet sample in Fig 8).

The second limitation is the low SNR of VLF imaging, affecting the attainable image resolution and recording time. Here we decided not to compensate SNR with averages, using the same number of excitations (NEX = 4) for all the setups, to limit the acquisition time. In this condition, since the SNR scales as $B_0^{7/4}$ in the low frequency coil-dominated noise regime, VLF imaging suffers from a poor SNR compared to higher fields. This limits, the spatial

resolution to a voxel size of 3 x 3 x 3 mm$^3$, since the SNR linearly scales with the voxel volume and higher spatial resolution would result in unacceptable acquisition time. This resolution allows 3D imaging but mis-shapes the sample geometry (the tube's section does not appear to be circular). Notably, for all the across-field comparisons shown in Figs 5 to 7, we used the same resolution also for the higher field images. This further affected the performance of the algorithms near, or on, the edges where we couldn't completely eliminate the partial volume effect due to instabilities in the R$_1$fitting because of low SNR. This reflects in the clustering instability of a few peripheral voxels even after edge erosion (see the few voxels which were incorrectly grouped in Fig 8), reducing the within-sample homogeneity and affecting the single-voxel sensitivity. This effect could in principle be reduced by increasing the SNR and reducing the voxel size, dramatically increasing the recording time. In the present work, the recording time was long but the sequence we used was not optimized for fast imaging, which still has to be implemented.

Especially for in-vivo measurements these limitations have multiple implications. The long recording times and the lower specificity for low T$_1$ times can be remedied by optimising the current sequence parameters or moving to other more efficient approaches like 2D multislice or alternative quantitative sequence designs [18, 19]. This however was impossible with the available hardware at the time. The lower SNR of VLF and thus the best spatial resolution, which is still suboptimal for in-vivo imaging, have further reaching restrictions, which are inherently fixed by the measurement field, limiting better single-voxel sensitivity and clustering accuracy. Improvements could be made by further optimising the existing hardware like the RF coils, amplifiers for lower noise floors, or the implementation of intermediate sensors to boost the signal strength [5]. Also, Artificial Intelligence techniques applied in the novel VLF systems operating at 50–60 mT [1, 15] should be tested at this lower field and SNR regime. We believe that a combination of these efforts should be promising to enable in-vivo measurements or at least boost the capability of VLF MRI making it a suitable alternative. It is important to note that even if with longer recording times and lower SNR and resolution, this technique could provide novel information on relaxation rates, which are not available with standard clinical setups at higher fields.

## Conclusion

The advantage of T$_1$ contrast in ULF MRI was already discussed some years ago for phantoms, showing T$_1$-weighted images of agarose gel [26, 41]. Projected 2D T$_1$ mapping demonstrated the potentiality of this technique in distinguishing different concentrations of MnCl$_2$, within a range of R$_1$ values similar to the one considered in this work [42]. Projected 2D T$_1$ mapping was also applied to different tissue types in ex-vivo prostate tumours and in the healthy brain [16, 17] 3D qMRI was then demonstrated in security applications, but with a voxel size considerably larger than in our case [43]. Our results extend these works, as we demonstrated the feasibility of 3D R$_1$ mapping at VLF with a sensitivity better than 2.7 s$^{-1}$ and a spatial resolution of 3 mm. This result is supported by statistical and clustering analysis showing that voxels within the samples were correctly grouped and distinguished, even for small CA concentration differences. This represents a clear advantage of 3D R$_1$ mapping compared to T$_1$-weighted images at VLF, since the latter did not produce reliable results and a robust distinction among different samples was not possible, especially for low CA concentrations. All these results are promising in the perspective of a factual application of 3D R$_1$ mapping at VLF for in vivo recordings.

Our research also provides information on 3D VLF MRI with Gd-based CA, complementing the findings on projected 2D T$_1$ mapping at ULF MRI through comparison with higher fields [42]. As shown in Figs 5 and 8, the lower CA concentrations were clearly separated and

distinguishable from the reference sample while at higher fields this was possible only for the higher CA concentrations. These results could be a good resource on applications of Gd-based CA at VLF in clinics, especially with deployment of recent 50–60 mT clinical systems (Swoop [15]). Of course, the future evolution of our findings will be VLF 3D quantitative mapping in other media and in tissues, to verify its effectiveness. The field-dependent behaviour of several CAs relaxometry parameters, showing $r_1$ enhancement at lower fields, can draw the wrong conclusion that effortless lesion contrast enhancement can be obtained by reducing the main magnetic field. This is not always true since CA relaxivity, as well as the tissue contrast before CA administration, both determine the final contrast after CA administration. At low fields, the former is often larger, but the latter is reduced by the inherently shorter tissue relaxation times.

Finally, it is worth noting that evidence is available of remarkable low-field performances of non-Gd-based CAs like Superparamagnetic Iron Oxide Nanoparticles (SPIONs, acting mostly on the transversal relaxation times). If the low-to-high field relaxivity ratio of Gd complexes generally is up to one order of magnitude, SPIONs can reach up to two orders of magnitude thus boosting the relaxivity effect on the final tissue contrast [44]. Thus, future studies should feature more CAs, still compared at different magnetic field strengths but use the same timing sequences wherever possible.

## Supporting information

**S1 File. Additional information.** Additional details on data pre-processing, working assumptions and ANOVA results.
(DOCX)

## Author Contributions

**Conceptualization:** Allegra Conti, Angelo Galante, Massimo Caulo, Stefano Sensi, Cosimo Del Gratta, Stefania Della Penna.

**Formal analysis:** Danilo de Iure, Sara Spadone, Stefania Della Penna.

**Funding acquisition:** Stefano Sensi, Stefania Della Penna.

**Investigation:** Allegra Conti.

**Methodology:** Danilo de Iure, Sara Spadone, Ingo Hilschenz, Stefania Della Penna.

**Software:** Danilo de Iure, Sara Spadone.

**Supervision:** Stefania Della Penna.

**Writing – original draft:** Danilo de Iure, Allegra Conti, Stefania Della Penna.

**Writing – review & editing:** Angelo Galante, Ingo Hilschenz, Massimo Caulo, Stefano Sensi, Cosimo Del Gratta.

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
