## [Decision Letter · Decision Letter 0]

14 Feb 2023

PONE-D-22-33727Analyzing the sensitivity of quantitative 3D MRI of longitudinal relaxation at Very Low Field in Gd-doped phantomsPLOS ONE

Dear Dr. De Iure,

Thank you for submitting your manuscript to PLOS ONE. After careful consideration, we feel that it has merit but does not fully meet PLOS ONE’s publication criteria as it currently stands. Therefore, we invite you to submit a revised version of the manuscript that addresses the points raised during the review process.

We look forward to receiving your revised manuscript.

Kind regards,

Marco Giannelli

Academic Editor

PLOS ONE

Journal Requirements:

Additional Editor Comments:

This is an interesting and original study. In order to further improve the manuscript, I hope the Authors are able and willing to adequately address some minor concerns raised by the Reviewers.

Reviewers' comments:

Reviewer's Responses to Questions

**Comments to the Author**

1. Is the manuscript technically sound, and do the data support the conclusions?

Reviewer #1: Yes

Reviewer #2: Yes

2. Has the statistical analysis been performed appropriately and rigorously? 

Reviewer #1: Yes

Reviewer #2: Yes

3. Have the authors made all data underlying the findings in their manuscript fully available?

Reviewer #1: Yes

Reviewer #2: Yes

4. Is the manuscript presented in an intelligible fashion and written in standard English?

Reviewer #1: Yes

Reviewer #2: Yes

5. Review Comments to the Author

Reviewer #1: The manuscript “Analyzing the sensitivity of quantitative 3D MRI of longitudinal relaxation at Very Low Field in Gd-doped phantoms” describes quantitative T1 measurement on a homemade Ultra low field magnetic resonance tomograph (8.9 mT). The work is original, and the topics addressed, ultra-low field MRI and quantitative T1 (or R1) mapping, are both very hot, and could potentially be of interest for many readers from different scientific fields. The experimental work was well conducted, and the paper is overall well written. I believe that this paper could have a certain impact. For all these reasons, I’m favorable to the work publication on PLOS ONE. I report below some minor issues to fix before pubblication.

1) Some more details about the MRI machine used could be useful: the gradients used (and slew/rate), should be reported.

2) The phantom used need a better description:

i) what is in the background? It is air?

ii) the overall phantom dimensions have not been reported

iii) probably a picture of the phantom could increase the readability.

3) page 9 line 174, the authors said “The latter condition (TR>T2* NDR) is met in our setup where the homogeneity of the measurement field is low, considerably reducing the effective T2*” but the T2* value is not reported. In my opinion the T2* should be reported (at least an estimate of this value).

4) Page 9 line 176- 177, an image showing the echo signal at the end of the acquisition could be helpful for supporting this claim.

5) Figure 4-5-6 the 2 digits after the dot (.00) can be omitted for increasing readability

Reviewer #2: This study tested the ability and the sensitivity of a VLF-MRI scanner operating at 8.9 mT in obtaining 3D longitudinal relaxation rate maps and distinguishing between voxels intensities. Phantoms consisting of

vessels doped with different Gd-based contrast agent concentrations were used. 3D R1 maps and T1-weighted MR images were analysed and results obtained at 8.9 mT (4 NSA and isotropic voxel size of 3 mm) were compared with commercial scanners operating at 0.2 T, 1.5 T, and 3 T.

The authors found that VLF R1 maps offered a higher sensitivity in distinguishing the different contrast agent concentrations and an improved contrast compared to higher fields. Moreover, the high sensitivity of VLF-MRI allowed an effective clustering of the 3D map values. Instead, T1-weighted images were less reliable, even at higher CA concentrations.

The discussed topic is specific but of particular interest, and the text is generally clear and well written.

Below some remarks:

- Correct some minor grammatical errors in the text (line 99 spurious "s", line 100 missed dot at the end, lines 104-105 missed comma before "respectively", line 408 close the parenthesis).

- Please provide a reference for the sentence at lines 118-119.

- At line 207 "subsection 3" does not clearly point to the corresponding subsection, that does not have a identifier number. Please specify better which is the reference subsection.

- Caption of Fig 7 (line 398): 8.9 mT, not 9 mT.

- Please stress more in the Discussion section the limits of low SNR and limited spatial resolution for the in vivo translation of this technique.

- Please provide the acquisition times for each sequence and scanner, to understand the possible translation in the in vivo field.

6. PLOS authors have the option to publish the peer review history of their article (what does this mean?). If published, this will include your full peer review and any attached files.

Reviewer #1: No

Reviewer #2: No

---

## [Author Response · Author response to Decision Letter 0]

31 Mar 2023

A detailed point-to-point reply can be found in the document file 'Response to Reviewers'.

I'm copying below the text of the reply (no figure allowed).

Please refer to the file 'Response to Reviewers' for the complete answer.

Dear Editor,

We thank the reviewers for their comments and suggestions, which contributed to increase the robustness of our results and improved the readability of the manuscript. We will address below each of the reviewers’ remarks.

Reviewer #1 

1) Some more details about the MRI machine used could be useful: the gradients used (and slew/rate), should be reported.

As suggested by the Reviewer, we added the relevant details on the MRI machines used in this work. Further details of our VLF MRI scanner can be found in the cited paper referred as [6] in the manuscript and, for the standard clinical systems, in the producers’ manual. To address this remark, we modified the text in the Recordings subsection as follow:

“MR images were recorded using four devices operating at different magnetic field strengths: a system implemented in the EU project MEGMRI, operating at 8.9 mT [6] , and three commercial systems used for clinical applications operating at 0.2 T (Esaote Artoscan C-Scan), 1.5 T and 3T (both Philips Achieva). The VLF system operated at a magnetic field of 8.9 mT generated by a compensated solenoid coil. A Maxwell coil generated a gradient field of 0.22 mT/(m�A) in the Z-direction (along the solenoid axis) and the X-Y gradient coils were designed using a Finite Element method and were located on the inner surface of the solenoid generating a gradient field of 0.38 mT/(m�A). The slew-rate was approximately 5 T/(m�s). The receiver coil Rx was a saddle style coil with a diameter of 8 cm and height of 6 cm. The transmission coil Tx was placed outside the Rx coil, rotated by 90° to achieve a decoupling of around 60 dB. The Artoscan was equipped with a permanent magnet to generate the 0.2 T field and field gradients of 10 mT/m with a slew rate of about 40 T/(m�s). This system was designed to image upper and lower limbs. The Philips Achieva systems were equipped with superconducting magnets and gradients of 33 mT/m and 40mT/m and slew rates 180T/(m�s) and 200 T/(m�s), respectively. ”

2) The phantom used need a better description:

 What is in the background? It is air?

 the overall phantom dimensions have not been reported.

 probably a picture of the phantom could increase the readability.

The phantom consisted of 5 vessels filled with about 1.5 ml of different dilutions of contrast agent placed in a piece of foam in a fixed relative position during all the measurements. No signal from the foam was noticeable with all the scanners. The foam was shaped to fit into the smaller receiving coil, i.e. the one of the 8.9 mT system. Fig 1, here presented, was added to the manuscript and properly addressed as suggested.

Fig 1: Upper left: An individual phantom vessel. Upper right: The 5 samples in the foam support, same positions as for the measurements. The samples are positioned as follows: 1- ref. sample, 2 – 0.17 mM, 3 – 0.25 mM, 4 – 0.5 mM, 5 – 1 mM. Each vessel is filled with about 1.5 ml of solution. Lower: The phantom is shown inside the VLF system.

3) page 9 line 174, the authors said “The latter condition (TR>T2* NDR) is met in our setup where the homogeneity of the measurement field is low, considerably reducing the effective T2*” but the T2* value is not reported. In my opinion the T2* should be reported (at least an estimate of this value).

In our experiments, during the acquisition window, the T2* is mainly driven by the gradient field. According to [1], field inhomogeneities affect the T2* as in the following: 

1/(T_2^* )=1/(T_2^* )+γΔB_inhom 

where T2 is the intrinsic transverse relaxation time, and ΔB_inhom is the magnetic field inhomogeneity across a voxel. To obtain an upper estimate of T2* we consider only the applied readout gradient discarding contributions from the intrinsic relaxation and the measurement field inhomogeneity. The applied sequence bandwidth is about 2.5 kHz, corresponding to 78 Hz for each of the 32 acquired voxels in the frequency direction. This results in a T_2^* "≤" "1" ⁄("γΔ" "B" _"inhom" ) "=13 ms" constraint, confirming the T_2^* "≪" T_R assumption (as the smallest TR used in our procedure was 85 ms). Further considerations are reported in the answer to the next issue. These considerations are added in the Methods section. 

4) Page 9 line 176- 177, an image showing the echo signal at the end of the acquisition could be helpful for supporting this claim.

As requested by the reviewer, we added a figure showing that at the end of the acquisition window in the spin-echo sequence used in the VLF system the signals were already degraded down to the noise floor level, meaning that no residual transverse magnetization was present at the beginning of a new repetition. The reading gradient is not compensated after the acquisition window, then it basically acts as a spoiling gradient for the reading direction and prevents coherence pathways effects when acquiring different lines of k-space.

In Fig 2 we show the averaged signal during the acquisition from the central lines of the k-space along the reading direction (in blue), corresponding to the smallest phase encoding gradients amplitude and the most intense signal. Conversely, the averaged signal (in red) during the acquisition from the peripheral lines of the k-space along the reading direction (corresponding to the most intense phase encoding gradient amplitudes and the smallest signal) is representative of the noise floor. At the end of the acquisition window, the amplitudes of the two traces are similar and, in particular, this holds true for the shortest TR (shown on the left), demonstrating that our assumptions on T_2^* "≪" T_R are always correct in our case.

No post-processing is applied to the signal and only an anti-aliasing filter (a Finite Impulse Response one) with a cut-off frequency equal to the Nyquist frequency (1.25 kHz) is acting during the acquisition.

For sake of clarity and to prevent a decrease of readability, we included Fig 2 and the relative description in the Supplementary Information and we added a reference to them in the text.

Fig 2. Raw signals and noise floor. Comparison between raw signals from k-space during the acquisition window at multiple TR. The shortest one is the most interesting, demonstrating that our assumption on T2* << TR is always correct in our case. The blue line represents the mean signal from the 16 central lines of the 3D k-space, i.e. with almost null encoding shift in the two phase directions, while the red line represents the average of 16 peripheral lines of the 3D k-space in both the phase encoding directions, assumed as representative of the noise floor. Signals are reported as acquired, with only a finite impulse response anti-aliasing filter and no post-processing.

5) Figure 4-5-6 the 2 digits after the dot (.00) can be omitted for increasing readability

Thanks for the remark, the labels on the axis of the figures were modified accordingly.

Reviewer #2 

1) Correct some minor grammatical errors in the text (line 99 spurious "s", line 100 missed dot at the end, lines 104-105 missed comma before "respectively", line 408 close the parenthesis).

Thanks for these remarks, the manuscript was modified accordingly. 

3) At line 207 "subsection 3" does not clearly point to the corresponding subsection, that does not have an identifier number. Please specify better which is the reference subsection.

We thank the reviewer for this remark, the typo was corrected.

4) Caption of Fig 7 (line 398): 8.9 mT, not 9 mT.

We corrected the typo.

2) Please provide a reference for the sentence at lines 118-119.

The doped water solution is a commercial product from Labochimica Srl (Padova, IT), it could be found on an online catalogue at this URL:

http://www.prodottichimicipadova.it/catalogo-prodotti/soluzioni-e-reagenti/rame-solfato-sol-sec-form-ml-1000

 In the manuscript we added the note “Commercial Product by Labochimica srl, ‘RAME SOLFATO Sol. Sec. Form. ml 1000’ ”.

5) Please stress more in the Discussion section the limits of low SNR and limited spatial resolution for the in vivo translation of this technique.

To address this remark, we added a few sentences after the end of the Discussion subchapter “Limitations of R1 mapping at VLF”. 

“The second limitation is the low SNR of VLF imaging, affecting the attainable image resolution and recording time. Here we decided not to compensate SNR with averages, using the same number of excitations (NEX= 4) for all the setups, to limit the acquisition time. In this condition, since the SNR scales as B07/4 in the low frequency coil-dominated noise regime, VLF imaging suffers from a poor SNR compared to higher fields. This limits, the spatial resolution to a voxel size of 3 x 3 x 3 mm3, since the SNR linearly scales with the voxel volume and higher spatial resolution would result in unacceptable acquisition time. This resolution allows 3D imaging but mis-shapes the sample geometry (the tube’s section does not appear to be circular). Notably, for all the across-field comparisons shown in Fig 5 to Fig 7, we used the same resolution also for the higher field images. This further affected the performance of the algorithms near, or on, the edges where we couldn’t completely eliminate the partial volume effect due to instabilities in the R1 fitting because of the low SNR. This reflects in the clustering instability of a few peripheral voxels even after edge erosion (see the few voxels which were incorrectly grouped in Fig 8), reducing the within-sample homogeneity and affecting the single-voxel sensitivity. This effect could in principle be reduced by increasing the SNR and reducing the voxel size, dramatically increasing the recording time. In the present work, the recording time was long but the sequence we used was not optimized for fast imaging, which still has to be implemented.

Especially for in-vivo measurements these limitations have multiple implications. The long recording times and the lower specificity for low T1 times can be remedied by optimising the current sequence parameters or moving to other more efficient approaches like 2D multislice or alternative quantitative sequence designs [18,19]. This however was impossible with the available hardware at the time. The lower SNR of VLF and thus the best spatial resolution, which is still suboptimal for in-vivo imaging, have further reaching restrictions, which are inherently fixed by the measurement field, limiting better single-voxel sensitivity and clustering accuracy. Improvements could be made by further optimising the existing hardware like the RF coils, amplifiers for lower noise floors, or the implementation of intermediate sensors to boost the signal strength [5]. Also, Artificial Intelligence techniques applied in the novel VLF systems operating at 50-60 mT [1,15] should be tested at this lower field and SNR regime. We believe that a combination of these efforts should be promising to enable in-vivo measurements or at least boost the capability of VLF MRI making it a suitable alternative. It is important to note that even if with longer recording times and lower SNR and resolution, this technique could provide novel information on relaxation rates, which are not available with standard clinical setups at higher fields.”

6) Please provide the acquisition times for each sequence and scanner, to understand the possible translation in the in vivo field.

We added the acquisition times, for NEX=4, for the minimum and the maximum TRs in Table 1.

Table 1. Addendum to MRI Acquisition parameters

 8.9 mT 0.2 T 1.5 T 3 T

Min/Max

Acquisition times [s] [348, …,2089]* [220, …, 540] [68, …, 408] [96, …, 612]

* Please note that the 8.9 mT 3D cartesian spin-echo is not optimized, since 32 slices have to be acquired, regardless of the phantom dimensions. Conversely, at higher fields 2D multislice sequences are used.

We acknowledge that in the current implementation the VLF instrument is far from being applied for in vivo studies, which would require further improvements of the hardware setup and the sequences. It is important to specify, tough, that the sequences used at the higher fields are standard spin-echo not meant for clinical quantitative estimation. Their acquisition times are not optimized or reduced, and comparing them could be tricky if not wrongful.

References:

 Chavhan, Govind B; Babyn, Paul S; Thomas, Bejoy; Shroff, Manohar M; Haacke, E. Mark (2009). "Principles, Techniques, and Applications of T2*-based MR Imaging and its Special Applications". Radiographics. 29 (5): 1433–1449. doi:10.1148/rg.295095034

We added the above reference on the main manuscript as [35] and rescaled the references list accordingly. We also escluded a spurious reference not cited in the manuscript (Huang , Tao , Chang , Dong. Field dependence study of commercial Gd chelates with SQUID detection. IEEE Transactions on Applied Superconductivity. 2016; 26(5): p. 1--4).

---

## [Decision Letter · Decision Letter 1]

24 Apr 2023

Analyzing the sensitivity of quantitative 3D MRI of longitudinal relaxation at Very Low Field in Gd-doped phantoms

PONE-D-22-33727R1

Dear Dr. De Iure,

We’re pleased to inform you that your manuscript has been judged scientifically suitable for publication and will be formally accepted for publication once it meets all outstanding technical requirements.

Kind regards,

Marco Giannelli

Academic Editor

PLOS ONE

Additional Editor Comments (optional):

I thank the Authors for the effort put to definitively improve this interesting manuscript.

Reviewers' comments:

Reviewer's Responses to Questions

**Comments to the Author**

1. If the authors have adequately addressed your comments raised in a previous round of review and you feel that this manuscript is now acceptable for publication, you may indicate that here to bypass the “Comments to the Author” section, enter your conflict of interest statement in the “Confidential to Editor” section, and submit your "Accept" recommendation.

Reviewer #1: All comments have been addressed

Reviewer #2: All comments have been addressed

2. Is the manuscript technically sound, and do the data support the conclusions?

Reviewer #1: Yes

Reviewer #2: Yes

3. Has the statistical analysis been performed appropriately and rigorously? 

Reviewer #1: Yes

Reviewer #2: Yes

4. Have the authors made all data underlying the findings in their manuscript fully available?

Reviewer #1: Yes

Reviewer #2: Yes

5. Is the manuscript presented in an intelligible fashion and written in standard English?

Reviewer #1: Yes

Reviewer #2: Yes

6. Review Comments to the Author

Reviewer #1: The Authors made a big effort to respond to all issues raised, the paper is in my opinion ready for publication.

Reviewer #2: The authors now addressed all my remarks in their updated version of the article, completely answering to the comments.

7. PLOS authors have the option to publish the peer review history of their article (what does this mean?). If published, this will include your full peer review and any attached files.

Reviewer #1: No

Reviewer #2: No

---

## [Editor Report · Acceptance letter]

28 Apr 2023

PONE-D-22-33727R1 

Analyzing the sensitivity of quantitative 3D MRI of longitudinal relaxation at Very Low Field in Gd-doped phantoms 

Dear Dr. de Iure:

I'm pleased to inform you that your manuscript has been deemed suitable for publication in PLOS ONE. Congratulations! Your manuscript is now with our production department. 

Kind regards, 

on behalf of

Dr. Marco Giannelli 

Academic Editor

PLOS ONE